# Impact of Different Layer Housing Systems on Eggshell Cuticle Quality and *Salmonella* Adherence in Table Eggs

**DOI:** 10.3390/foods10112559

**Published:** 2021-10-23

**Authors:** Garima Kulshreshtha, Cristina Benavides-Reyes, Alejandro B. Rodriguez-Navarro, Ty Diep, Maxwell T. Hincke

**Affiliations:** 1Department of Cellular and Molecular Medicine, Faculty of Medicine, University of Ottawa, Ottawa, ON K1H 8M5, Canada; gkulshre@uottawa.ca; 2Departamento de Mineralogia y Petrologia, Campus de Fuentenueva, Universidad de Granada, 18002 Granada, Spain; crisbr@ugr.es (C.B.-R.); anava@ugr.es (A.B.R.-N.); 3Lyn Egg Production and Grading, Burnbrae Farms Limited, Lyn, ON K0E 1M0, Canada; tdiep@burnbraefarms.com; 4Department of Innovation in Medical Education, Faculty of Medicine, University of Ottawa, Ottawa, ON K1H 8M5, Canada

**Keywords:** eggshell cuticle, food safety, caging system, antimicrobial, *Salmonella*

## Abstract

The bacterial load on the eggshell surface is a key factor in predicting the bacterial penetration and contamination of the egg interior. The eggshell cuticle is the first line of defense against vertical penetration by microbial food-borne pathogens such as *Salmonella* Enteritidis. Egg producers are increasingly introducing alternative caging systems into their production chain as animal welfare concerns become of greater relevance to today’s consumer. Stress that is introduced by hen aggression and modified nesting behavior in furnished cages can alter the physiology of egg formation and affect the cuticle deposition/quality. The goal of this study was to determine the impact of caging systems (conventional, enriched, free-run, and free-range), on eggshell cuticle parameters and the eggshell bacterial load. The cuticle plug thickness and pore length were higher in the free-range eggs as compared to conventional eggs. The eggshells from alternative caging (enriched and free-range) had a higher total cuticle as compared to conventional cages. A reduction in bacterial cell counts was observed on eggshells that were obtained from free-range eggs as compared to the enriched systems. An inverse correlation between the contact angle and *Salmonella* adherence was observed. These results indicate that the housing systems of layer hens can modify the cuticle quality and thereby impact bacterial adherence and food safety.

## 1. Introduction

Table eggs are a readily available and inexpensive source of protein in the human diet. Annually, the Canadian egg industry produces approximately 10 billion eggs. Egg production rose 5.3% to 71.0 million dozen eggs from March 2019 to March 2020 [1]. In the United States, about 97 billion table eggs were produced in 2020, with an average production of 286 eggs/layer hen [2]. Globally, conventional cages are the most common production system for table eggs; however, due to hen welfare concerns, many countries have supplemented/replaced conventional cages with alternative systems [3,4]. In recent years, consumers, egg producers, legislators, consumer groups, as well as animal welfare organizations have shown an increased interest in the retail egg production system [3]. An analysis of a report on caging systems (2007 to 2021), that was published by the United States Department of Agriculture (USDA), showed that 29.2% of all table eggs are produced by laying hens that are housed in cage-free systems [5]. In Canada, about 71% of eggs are laid by hens that are housed in conventional or battery wire cages with a stocking density of 67–75 square inches/hen. The remaining 29% are housed in either enriched, free-run, or free-range systems [6]. Enriched cages provide more space for each hen (90 square inches) and are equipped with a perch, nest, and a scratching area. The free-run system represents an open concept barn with a variety of nests and perches; moreover, hens in free-range systems have access to the outdoors [7]. There is a significant trade-off with respect to hen welfare in caged vs. cage-free housing systems. Battery cages are associated with good health, superior hygiene, and low mortality, but restrict normal behaviors such as movement, stretching, and wing-flapping [8]. On the other hand, hens that are housed in cage-free systems have better opportunities to express normal behaviors but face higher risks of mortality, injury, and poorer health [8,9].

Legislation in Europe that came into effect in 2012 banned the use of conventional cages for laying hens. Following consumer concerns about conventional cages, only the enriched cage or non-cage systems are currently allowed in the European Union [10]. In Canada and the United States, changes to hen housing systems are being implemented at a slower pace and are mainly market- or producer-driven in response to pressure from customer and animal protection groups, rather than prohibition by government regulation and policies [11]. In February 2016, Egg Farmers of Canada announced a policy forbidding new conventional cage systems after July 2016. Moreover, the Retail Council of Canada is committed to providing only cage-free eggs by the end of 2025 [6,8,12]. These systems are designed to allow birds to express a fuller repertoire of normal behaviors due to opportunities for interaction with their environment and other birds [9]. As the industry transitions away from conventional cages, the optimization of farm management and layer genetics will be required to achieve maximum efficiency from alternative housing systems.

A pathogen-free egg is extremely important for the food safety of the nutritious unfertilized egg; food safety-associated recalls of eggs due to contamination with bacterial pathogens, especially *Salmonella*, is a public health concern [13,14]. In addition to economic losses to the poultry industry, negative publicity reduces consumer confidence in this nutritious food [14,15]. In Canada and the United States, eggs are graded/washed before retail sale. This practice removes potential pathogens from the eggshell surface and protects consumers from undue risk [12]. Nonetheless, bacterial pathogens continue to be introduced into the food chain from contaminated eggs [16]. Globally, it is estimated that over 80.3 million illnesses occur due to foodborne *Salmonella* infection, with contaminated poultry being the largest contributor to salmonellosis [17]. In the United States, approximately 1.35 million cases of salmonellosis (caused by non-typhoidal *Salmonella*) are reported every year. In addition, the CDC estimates that *Salmonella* poisonings cause approximately 27,000 hospitalizations and 420 deaths in the United States every year (CDC, 2021) and an economic burden of 2.8 billion USD/year [18]. In Canada, it is estimated that about 6000–12,000 annual cases of *Salmonella* poisoning are due to contaminated poultry products [19].

The eggshell and its cuticle are the first line of defense against adhering and penetrating microorganisms, providing both protection from pathogens and a robust container for the egg’s contents [20,21]. Microbial pathogens may enter the egg through respiratory pores that perforate the eggshell [22]. However, the outer openings of the respiratory pores are usually covered by a proteinaceous cuticle, which extends into pores up to 50 μm and restricts the entry of bacterial pathogens [23]. An intact cuticle is a physical barrier covering the eggshell, which is critical for food safety against pathogens such as *Escherichia coli* and *Salmonella* [24,25,26]. Our recent study revealed that cuticle composition and chemistry are influenced by hen age and strain, and by the commercial washing process. Scanning electron microscopy (SEM) observations confirmed that the commercial washing process can wear away some of the outer surface cuticle, however, the protein that acts as the cuticle plug is still intact and would continue to block bacteria from transiting the respiratory pore [27]. Eggs can become contaminated by pathogens at different stages including production, processing, preparation, and consumption. There are two major routes of microbial contamination of eggs: vertical- and horizontal-transmission. Vertical transmission is trans-ovarian and occurs during egg formation, while horizontal transmission occurs after lay through penetration of the eggshell [28]. The microbial contamination of eggshells progresses in different stages: bacterial attachment to the cuticle and shell, penetration through cracks or respiratory pores, followed by colonization of the underlying membranes, and contamination of the albumen and yolk resulting in infection of egg contents [29].

The chemical constituents of eggshell cuticles such as glycoproteins, polysaccharides, and lipids function as a chemical barrier to prevent pathogen invasion and limit bacterial contamination of the egg contents [20,30,31,32,33,34]. Diminished cuticle coverage, for example, due to washing of eggs, favors the loss of moisture and an increased susceptibility to bacterial contamination from the exposed respiratory pores and a reduction in surface antimicrobial proteins. The cuticle coverage is traditionally estimated by the degree of staining of the eggshell with MST (MS Technologies, Northamptonshire, UK) cuticle blue stain [26,35,36]. A recent study used optical theory to improve the staining method to evaluate the cuticle quality of different colored eggs [37]. Another method based on Attenuated Total Reflection- Fourier Transform Infrared spectroscopy (ATR-FTIR) can characterize both the cuticle quality (i.e., the thickness/degree of coverage over the eggshell mineral) as well as the chemical composition of the surface. The ultrastructure (spherical vesicles of different sizes: 50 to 500 nm) and the morphology of the cuticle can also be evaluated by scanning and transmission electron microscopy [38]. 

It has been demonstrated that hen aggression and nesting behavior are modified in furnished or alternative cages [39]. These factors can modify the physiology of egg formation and cuticle deposition/quality since environmental stress due to changes in housing or proximity of unfamiliar birds can delay oviposition [40]. Moreover, cuticle deposition is associated with the normal endocrine cycle, which can be altered by environmental stress such as pen to cage transfer [41]. Previous research has demonstrated an effect of the housing system on egg contamination by bacteria [42,43,44]. However, the impact of the caging system on the eggshell cuticle quality and cuticle pore plugs, which are critical to exclude bacteria from the egg surface and interior, has not yet been studied. Therefore, in this study, we have compared the chemical composition of the eggshell cuticle and structure of pore plugs of eggs from four different housing systems: conventional, enriched, free-run, and free-range. Additionally, the influence of housing systems on the surface hydrophobicity and *Salmonella* adherence was evaluated, as we hypothesized that these parameters play an important role in maintaining food safety of table eggs.

## 2. Materials and Methods

### 2.1. Source of Eggs and Description of Housing Systems

Eggs from four different housing systems were provided by Burnbrae Farms Limited (Lyn, ON, Canada): conventional (wired battery cages, which are about 67–75 square inches/hen); enriched cages (birds housed in large groups furnished with nest space (6.3 square inches); perches (4.4 inches) and scratch mat (3.8 square inches; free-run (cage-free indoor environment); and free-range (cage-free environment with access to the outdoors). All of the eggs that were used in this study were unwashed eggs (*n* = 60) from Lohmann white or brown hens at the mid-lay (44–48 weeks of age). 

### 2.2. Chemicals and Instruments Used in This Study

A total of 30 eggs were randomly selected from each housing system for cuticle staining, ATR-FTIR and surface hydrophobicity analysis. Of these, 6 eggs were randomly selected from each group for SEM and energy-dispersive X-ray spectroscopy (EDS), and another 18 eggs were used for bacterial cell attachment assay.

All of the eggs were visually inspected for cracks, dirt, feathers, and pinholes (*n* = 18–20 out of 60 eggs), and only clean and intact eggs were included in the analyses. The internal contents (albumen and yolk) of each egg were cleaned using a standardized method described previously [27]. Care was taken to avoid washing the outer surface of the egg during the cleaning process. The processed eggshells were stored at −20 °C prior to the experimental analysis. All of the eggshells were randomized before analysis to avoid any bias. MST cuticle blue was purchased from MS Technologies Ltd., Northamptonshire, UK. All of the other chemicals that were used in this study, unless indicated otherwise, were purchased from Sigma Aldrich (Oakville, ON, Canada) or Thermo Fisher Scientific (Waltham, MA, USA).

The eggshells that were treated with bleach to remove the cuticle were used as a control in this study. The cuticle was removed from the outer surface of the eggshells by incubating intact eggs in a 5% solution of bleach (sodium hypochlorite, LavoPro 6, Montreal, QC, Canada) for 10 min (repeated up to 6 times, depending on the degree of cuticle removal). We confirmed that the cuticle was removed by MST cuticle blue staining. 

### 2.3. Visualization of Eggshell Pore Plug Parameters: Scanning Electron Microscopy and Elemental Analysis (EDS)

The cuticle pore plugs were visualized on cross-fractured eggshell fragments from the blunt end of each egg (*n* = 3/fragment, 10–15 mm^2^) using a scanning electron microscopy (SEM; TeScan Vega-II XMU, Brno—Kohoutovice, Czech Republic) at 20 kV. All samples were carbon-coated at the Nano Imaging Facility, Carleton University, Ottawa, ON, Canada. ImageJ software bundled with java 1.8.0_172 (U. S. National Institutes of Health, Bethesda, MD, USA) for analysis of images was used to measure pore length, pore width (diameter of mouth of respiratory pore), pore plug thickness (depth/length of cuticle plug), pore length and eggshell thickness. Elemental analysis was carried out using energy dispersive X-ray spectroscopy (EDS, INCA Energy 250, Oxford Instruments, Abingdon-on-Thames, UK). For each hen housing system, a total of 18 cuticle pore plugs from 6 eggs were analyzed.

### 2.4. Cuticle Measurement: Cuticle Staining and Infrared Spectroscopy Analysis

#### 2.4.1. Cuticle Staining with MST Cuticle Blue

The quality and the degree of coverage of the eggshell cuticle was determined using MST cuticle blue staining as described previously [38]. The eggshell fragments (*n* = 30, 1 cm^2^) from either the blunt pole or the equatorial region were immersed in a solution of 1% MST cuticle blue dye in water for 5 min to stain the outer surface cuticle proteins, followed by rinsing in sterile Milli-Q water to remove excess surface dye before drying at room temperature for 24 h. The cuticle coverage was determined by the intensity of green cuticle staining on the eggshell surface. The “L*, a*, and b*” color space values, measuring the degree of lightness, L*, and the chromaticity coordinates, a* and b*, were quantified at three randomly selected zones of each eggshell surface using a Konica Minolta spectrophotometer (CM-2600d). The average of L*, a*, and b* values before (randomly selected) and after staining were calculated per eggshell fragment and used to calculate Δ*E*_ab_*. A higher Δ*E*_ab_* equals a higher staining intensity
$$\Delta E_{ab}^{*}=\surd [(\Delta L^{*2})+\left(\Delta a^{*2}\right)+\left(\Delta b^{*2}\right)]$$

#### 2.4.2. Attenuated Total Reflection Fourier Transform Infrared Spectroscopy (ATR-FTIR)

The chemical composition of the cuticle of each eggshell fragment was determined by ATR-FTIR as described previously [38]. The outer surface of each eggshell fragment was pressed against the ATR diamond crystal window (Pro ONE, JASCO, Hachioji, Tokyo, Japan), and the IR spectra were recorded using an FTIR spectrometer (model 6200, JASCO Analytical Instruments, Hachioji, Tokyo, Japan). The amounts of proteins, sulfate, carbonate, sugar/polysaccharide, and water were calculated using the area of absorption peaks that were associated with molecular groups that were specific for each component. The total cuticle was represented as the sum of the signals from the protein constituents: amides; sulfates, phosphates, and sugars/polysaccharides [38]. For each hen housing system, a total of 30 eggs were analyzed.

Pearson correlation analysis for all IR peaks was used to establish a relationship between the main chemical components of the cuticle and surface hydrophobicity and the bacterial adherence.

### 2.5. Measurement of Surface Hydrophobicity—Contact Angle

The surface hydrophobicity (contact angle) for the eggshell surfaces (*n* = 30) was analyzed by the contact angle goniometer (Dataphysics OCA 15EC, San Jose, CA, USA) sessile drop technique (Appendix A). Deionized water (4 µL) was dropped onto the eggshell surface at a dosing rate of 2 µL/s. The acquired images were used to measure the contact angle of the droplet using the SCA20 software (Dataphysics). For each hen housing system, a total of 30 eggs were analyzed.

### 2.6. Bacterial Cell Attachment Assay

#### 2.6.1. Bacterial Strain and Growth Condition

The antibiotic resistant *Salmonella enterica* subsp. Enterica serovar Typhimurium str. SL1344 was obtained from Dr. Subash Sad and Dr. Ryan Russell (Faculty of Medicine University of Ottawa). Luria–Bertani (LB) agar or broth (BioShop, Burlington, ON, Canada) containing ampicillin (100 μg/mL) was used for the maintenance and growth of bacterial cultures. The bacterial cultures were revived by streaking the glycerol stocks on LB agar plates followed by incubation at 37 °C for 24 h. Single colonies from LB agar were incubated in LB broth (3 mL) overnight at 37 °C, 250 rpm. The next day, the inocula were diluted 1:50 in fresh LB broth and grown until log phase (optical density = 0.2; ∼10^8^ CFU ml^−1^) at 600 nm. The bacterial suspension was pelleted at 13,000× *g*, 4 °C for 10 min, washed with Dulbecco’s phosphate-buffered saline (PBS) three times, and resuspended in PBS at 0.2 OD.

#### 2.6.2. Determination of *Salmonella* Adherence

A cell attachment assay was developed to evaluate the adherence of *Salmonella* Typhimurium to the eggshell surface. Briefly, a *Salmonella* Typhimurium suspension in PBS (50 μL, 0.2 = OD, 10^7^–10^8^ cells/mL) was placed onto a 1 cm^2^ eggshell fragment (*n* = 18/housing system) before incubation in a humid chamber for 3 h at 37 °C (Appendix A). After incubation, the shell pieces were gently rinsed with PBS to remove excess/non-adhering *Salmonella* cells. Each shell fragment was then vortexed in PBS (2 mL, 3 pulses-5 sec each) to dislodge firmly adhering *Salmonella* cells. Serial dilutions were plated on LB agar for overnight growth to determine the cell numbers for calculation of adherent bacterial density per cm^2^ of shell surface. Each eggshell fragment was imaged to calculate the areas of droplet and shell fragment using ImageJ software. 

The bleach treated eggshells were used to optimize this assay. Sonication (1 cycle 5 min, room temperature, Ultra-Sonicator, Fisher Scientific), and vortexing (3 pulses-5 sec each) were compared to dislodge bacterial cells. The vortexing was the most efficient (Appendix A) and was used in an optimized protocol to dislodge adhering *Salmonella* cells. Additionally, serially diluted bacterial doses (10^−1^, 10^−2^, 10^−3^, and 10^−4^ cells/mL) were applied on the bleach treated eggshell surface to determine the dose-dependence of the adhering *Salmonella* cells. The bacterial dose that was applied on the eggshell surface was plotted against adhering cells to evaluate the linearity and to validate this protocol (Appendix A).

### 2.7. Statistical Analysis

All of the statistical analyses were performed using the software package MINITAB 17 (Minitab, LLC, State College, PA, USA) and R Statistical Software (version 3.2.4; R Foundation for Statistical Computing, R Development Core Team, Vienna, Austria). The data showing a normal distribution were compared using ANOVA. A one-way ANOVA was used to compare the differences between the housing systems and a Student’s *t*-test was used to compare the differences between the hen strains (white vs. brown). Tukey’s HSD, as a post hoc test, was used to compare the differences between the least square means. The standard deviation (SD) was reported with the mean values. The significance level chosen for all analyses was *p* < 0.05.

## 3. Results

### 3.1. Visualization of Cuticle Pore Plugs 

The ultrastructure and morphology of the cuticle pore plugs were visualized using a SEM. The pore plugs in the eggshells from all of the housing systems were cone-shaped to conform to the funnel shape of the mouth of each respiratory pore (Figure 1A). Image analysis revealed that the pore plugs in the shells from the free-range eggs were significantly deeper/longer (34.74 ± 7.50 μm) than those from the conventional system (27.52 ± 4.38 μm) (*p* = 0.01, *n* = 6 × 3 = 18) (Figure 1C). The cuticle pore length and the shell thickness of eggs from the alternative caging system were higher (*p* = 0.0001 and 0.000, respectively, *n* = 18) compared to those from the conventional cages (Figure 1D,E). There were no significant interactions between the bird housing systems and the pore width (diameter of mouth of respiratory pore).

The hen strain had a significant effect on the pore length and eggshell thickness. The cuticle pore length and eggshell thickness that were visualized in brown eggs were significantly higher (*p* = 0.000 and 0.0002, respectively, *n* = 18) than in the white eggs (Table 1).

EDS analysis of the cuticle plugs and the eggshell mineral content validated our previous findings [27] that the chemical composition of the inner surface of the pore plug was consistently different from that of the surrounding eggshell mineral composition in eggs from different housing systems. No interactions were observed among the pore plug composition for major elements such as calcium, phosphorous, sulfur, and magnesium, with the bird housing system or bird strain (Appendix A). However, a trend of higher sulfur and magnesium content was observed in the pore plugs of enriched and free-range eggs, as compared to those from conventional and free-run systems (Appendix A). Additionally, a trend of higher sodium and potassium was observed in the pore plugs of free-range eggs as compared to eggs from other housing systems (Appendix A).

### 3.2. Cuticle Chemical Composition

The eggshell surface was analyzed by ATR-FTIR to determine the chemical composition of the cuticle. The main components that were identified in the FTIR spectra comprised a broad band from OH with amide A from residual water and proteins (3700 to 2500 cm^−1^), amide I and amide II proteins (1633 cm^−1^ and 1540 cm^−1^), carbonate from eggshell calcite (1390 cm^−1^), sugars representing polysaccharides (1100 to 990 cm^−1^), and sulfates (1240 cm^−1^) (Appendix A). The peaks for proteins (amide) and polysaccharides could be structurally associated as glycoproteins [38]. As described in our previous study, eggshells with a good cuticle quality showed high intensity of the amide (protein) and polysaccharide bands, and a low intensity of the carbonate band [27,38]. 

#### 3.2.1. Effect of Cuticle Removal by Bleach Treatment

FTIR analysis showed that the bleach treatment significantly (*p* = 0.0001, *n* = 30) removed the outer surface cuticle from the eggshells. There was a significant reduction in the intensity of amide I (proteins) in the bleach-treated eggs (0.0012 ± 0.0018, *n* = 30) as compared to the eggs from different housing systems (0.0344 ± 0.0197, *n* = 120). Upon reduction in the cuticle coverage or thickness, the carbonate peaks intensity increased as the underlying eggshell minerals became surface-exposed and were detectable. Higher (*p* = 0.0001) carbonate signal was observed in the eggshells from bleach treated eggs (0.2815 ± 0.0250, *n* = 30) than the eggshells from the different housing systems (0.1560 ± 0.0521, *n* = 120), supporting that the bleach removes most of the cuticle.

#### 3.2.2. Effect of Bird Housing Systems

Enriched and free-range eggs had a significantly higher (*p* = 0.0001, *n* = 30) cuticle compared to the free-run eggs (Figure 2A). Similar differences in the degree of staining were observed using MST cuticle blue dye. The free-run eggs had significantly lower ΔE*_ab_ values than the free-range and enriched eggs (Figure 3). The carbonates and sulfates were higher (*p* = 0.0001, *n* = 30) in the free-run than the enriched eggs (Figure 2B,D), while the sugars, representing polysaccharides and glycoproteins, were higher in the enriched than the free-run eggs. The interaction between the peaks that were associated with lipids and the housing systems was not significant (*p* > 0.05, *n* = 30). As determined by ATR-FTIR, the cuticle composition of eggs from conventional cages was not significantly different from those from alternative housing systems.

#### 3.2.3. Effect of Bird Strain

Brown eggs had significantly more cuticle amides (proteins, *p* = 0.012) and less sugars (polysaccharides, *p* = 0.02) than the white eggs (Figure 4C,D). No differences were observed in the intensities of the total cuticle and carbonates. However, a trend of more cuticle and less carbonate was observed in the brown compared to the white eggs (Figure 4A,B).

#### 3.2.4. Correlation between Chemical Components of the Cuticle 

The cuticle protein parameters including total cuticle and OH-Amide A signals were negatively correlated with the mineral carbonate signal (R = −0.9681, and R = −0.9560, *p* = 0.00001 respectively, *n* = 30) and positively correlated with sugars (R = 0.9186, *p* = 0.00008 and R = 0.6606, *p* = 0.00001 respectively, *n* = 30) and phosphates (R = 0.5127, *p* = 0.0019 and R = 0.3383, *p* = 0.0002) (Figure 2E and Appendix A). A strong positive correlation was observed between sulfate and carbonate (R = 0.9227, *p* = 0.00001) (Figure 2E and Appendix A). The strong positive correlation among the cuticle chemical components (e.g., proteins and sugars) reflects the presence of glycoproteins.

### 3.3. Hydrophobicity of Eggshell Surface

#### 3.3.1. Effect of Cuticle Removal by Bleach Treatment

The contact angle on the eggshell surface with a normal cuticle coverage (CA = 90–106°, *n* = 120) was higher (*p* = 0.0001, *n* = 30) as compared to the bleach treated eggs was below 90° (CA = 27–58°, *n* = 30) (Figure 5). By definition, a surface with a CA > 90° is hydrophobic, while CA < 90° is a hydrophilic surface. Thus, the cuticle components render the eggshell surface hydrophobic. 

#### 3.3.2. Effect of Bird Housing Systems

The contact angle of the surface of the enriched eggs was significantly lower (*p* = 0.0001, *n* = 30) than the eggs that were obtained from conventional, free-run, and free-range systems (Figure 5).

#### 3.3.3. Effect of Bird Strain

Although a trend of higher contact angles were observed in the white (99.67 ± 13.33°) compared to brown eggs (95.74 ± 7.37°), this difference was not significant (*p* = 0.076). 

#### 3.3.4. Correlation between Contact Angle and Cuticle Chemical Composition

A moderate positive correlation (*p* = 0.0001) was observed between the contact angle and the cuticle protein parameters, including OH-Amide A (R = 0.6043) and amides I and II (R = 0.5069), while there was a moderate negative correlation of the CA (*p* = 0.0001) with the ES mineral parameters: carbonates (R = −0.5860) and sulfates (R = −0.5550) (Figure 6).

#### 3.3.5. Correlation between Contact Angle and MST Cuticle Staining

No significant correlation (R = 0.0321; *p* = 0.8662) was observed between the contact angle and the cuticle estimation by MST cuticle blue.

### 3.4. Bacterial Cell Attachment Assay

Our preliminarily results during the optimization of this assay showed that vortexing (3 pulses-5 sec each) was more efficient (*p* = 0.0031) in dislodging the *Salmonella* cells as compared to vortex + sonication (5 min) (Appendix A). This method was validated by the strong correlation (R^2^ = 0.9453) between adhering *Salmonella* cell counts and serially diluted bacterial doses that were applied to the eggshell surface (Appendix A). 

The bacterial cell attachment assay showed that the removal of cuticle using bleach treatment significantly increased (*p* = 0.0001, *n* = 18) the numbers of adhering *Salmonella* Typhimurium on the outer surface of the eggshells, as compared to eggshells with an intact cuticle across all of the housing systems (Figure 7A). 

A reduction (*p* = 0.025) in the bacterial cell counts was observed in the eggs that were obtained from free-range as compared to the enriched systems (Figure 7B). No difference (*p* = 0.829) in the bacterial adherence was observed between the conventional and alternative caging systems. The brown eggs had higher (*p* = 0.01) adhering *Salmonella* cells as compared to the white eggs (Figure 7B). A strong negative correlation of bacterial cell count (on eggs from all housing systems) was observed with contact angle (R = −0.8201, *p* = 0.0001) (Appendix A). 

## 4. Discussion

The eggshell cuticle is a proteinaceous layer which regulates gaseous exchange and functions as a first line of defense against invading pathogens such as *Salmonella* [24,26,35,45,46]. The cuticle deposition is influenced by a number of factors including the age, strain, and health of the laying hen [27,38,47]. It is well documented that the eggshell quality is a labile trait which is susceptible to change in response to various housing systems [7,48,49,50,51,52], however, not much is known about the impact of the housing system on cuticle deposition. Here, we compared the chemical composition of the eggshell cuticle, the structure of pore plugs, and the surface hydrophobicity in table eggs from different housing systems. Additionally, the influence of housing system on *Salmonella* adherence in table eggs was determined.

The SEM showed that pores were longer, and the plugs were thicker in the free-range as compared to the conventional eggs (Figure 1). This could be due to a lower weight of eggs in conventional as compared to free-range systems. The size-related parameters (diameter, thickness, and surface area) of pore plugs have been previously shown to positively correlate with the egg weight, size, and pore number [27,53,54]. The eggshell thickness was directly correlated with the pore length (R = 0.6837, *p* = 0.0017). A distinct hen environment in each housing system could affect hen metabolism and the eggshell fabrication and its resulting microstructure, leading to differences in the eggshell thickness and strength [52,55]. The eggshells were thicker in eggs from free-range systems compared to the enriched and conventional systems. Similarly, previous studies also documented greater eggshell thickness in eggs from non-cage systems and outdoor layers as compared to the caged conventional system [7,49,50,51,56]. In contrast, some studies demonstrated a higher shell thickness in eggs from cages as compared to the cage-free systems, while others showed no interaction between the housing systems and eggshell thickness [48,57]. These different outcomes between studies might be due to the interaction of genotype and housing system or to differences in the age of the laying hens. 

A higher eggshell thickness in the free-range eggs could be due to differences in strain of birds: brown vs. white. Previous studies have shown thicker shells in brown as compared to white eggs [58,59]. Within white eggs, the shell thickness in eggs from alternative cages was higher than conventional cages. This could be related to calcium (Ca) metabolism, as shell quality assessment parameters are a common indicator of Ca metabolism in layers [60,61]. Maintaining the optimal Ca and phosphorus content in feed is critical to optimize the egg quality parameters in housing systems [55]. 

Eggs from alternative caging systems (enriched and free-range) had higher total cuticle and lower carbonate signal, by ATR-FTIR, as compared to conventional cages (Figure 2). Similarly, cuticle staining showed a trend of lower cuticle coverage in eggs from the conventional cages as compared to the enriched and free-range cages (Figure 3). This is in agreement with a previous study where the cuticle coverage was higher in free-range compared to cage eggs [47]. The cuticle quality on the eggshell surface can be influenced by a range of factors including hen stress [35,41]. Cuticle deposition requires a normal endocrine cascade and is susceptible to environmental stressors [40,41]. Birds that are under stress release the hormone adrenaline which causes a delay in oviposition and terminates cuticle deposition in the shell gland [35,40,62]. Changes in the housing system and proximity of unfamiliar birds have been shown to reduce cuticle deposition [40,41]. Heterophil to lymphocyte ratio (H/L ratio) is a reliable welfare marker for bird stress in different housing conditions [63]. A higher H/L ratio in hens has been observed in conventional systems as compared to enriched and free-range systems [7,64]. Hence, it could be inferred that the higher total cuticle in the non-cage system as compared to the conventional system could reflect lower levels of bird stress. 

A correlation analysis between AFT-FTIR peak areas revealed a relationship between the chemical components of the cuticle (Figure 2). Specifically, the peaks that were associated with proteins (total cuticle, amide, OH-Amide A) were strongly negatively correlated with carbonates. This indicates that with lower a cuticle coverage or a thinner cuticle, the protein coating on the eggshell surface decreased, exposing the eggshell carbonate mineral for its detection by ATR-FTIR (the penetration depth of the signal is typically 0.5–2 μm). The proteins were moderately correlated with the sugars and phosphates (phosphoproteins) indicating a gradient in the chemical composition of the cuticle with the outer part being richer in proteins and the inner part being abundant in polysaccharides and phosphates (Figure 8). The chicken egg cuticle is structurally composed of two layers, an organic-rich outer layer and an inner mineralized layer containing hydroxyapatite crystals [65]. Although most of the phosphate in the cuticle is in a non-crystalline form such as phosphoproteins (i.e., Ovocalyxin-32 and Osteopontin) [66,67], crystalline phosphate (hydroxyapatite) could also contribute to presence of phosphate in the inner cuticle.

The removal of the organic cuticle by bleach treatment resulted in a hydrophilic eggshell surface, suggesting that the cuticle proteins are responsible for the hydrophobicity of the outer eggshell surface (Figure 5). Previously it has been shown that the cuticle protein components, including Ovocleidin-17, are destroyed by bleach treatment [68]. In our measurements, the contact angle was positively correlated with the cuticle components and negatively correlated with the carbonate signal (Figure 6). This indicates that hydrophobicity can be used as a measure of the cuticle depth/completeness or “quality” in table eggs. Moreover, this non-destructive method could also be implemented for the estimation of the cuticle quality in hatchery eggs without compromising the embryonic development, which is appealing as this measurement can be easily incorporated into breeding programs to improve the biosecurity of eggs [69]. This approach warrants further investigation to evaluate a correlation between the contact angle and the total cuticle in fertilized eggs. The hydrophobic nature of the cuticle prevents wetting and water entry through the eggshell surface to resist bacterial infection [70,71]. It is well documented that the surface wettability has an impact on the primary bacterial adhesion [72,73]. An inverse correlation between the eggshell contact angle and *Salmonella* adherence has been established in this study (Appendix A), indicating that surface hydrophobicity, due to the presence of cuticle proteins, can reduce bacterial contamination.

The bacterial load on the eggshell surface is a key factor in predicting the bacterial penetration and contamination of the egg interior [22,28]. We developed a bacterial cell attachment assay to evaluate the role of the eggshell cuticle on the bacterial adherence. All of the eggshells that were from different housing systems exhibited low adhering *Salmonella* counts when they were incubated with a standardized bacterial solution under these defined conditions, and the number of attached bacteria increased significantly when the cuticle was removed by bleach treatment (Figure 7A). This verifies that the cuticle functions to reduce the bacterial adherence in addition to functioning as a barrier to microbial migration through the respiratory pores as described in previous studies [74,75]. The absence of the eggshell cuticle increases bacterial contamination due to the exposed openings of the pore plugs and reduced surface antimicrobials [76]. A number of antimicrobial proteins such as lysozyme, Ovotransferrin, Ovocalyxin-25, Ovocalyxin-32, Ovocalyxin-36, cystatin, Ovoinhibitor, vitellogenin-2, and Ovostatin have been identified in the proteomic analysis of the chicken egg cuticle [20,26,30,31]. Among these, Ovotransferrin inhibits the growth of Gram-negative *Salmonella* by binding and sequestering iron (Fe^3+^), which is an essential nutrient [21,77]. The adherence of bacteria to host cells, or to inanimate surfaces, is mediated by adhesins, which recognize and bind to specific receptors on the host cell surfaces [78,79]. Glycoproteins function as receptor analogs and block specific bacterial adhesion ligands in many pathogens [78,80]. For example, sialylated fractions of the egg yolk, sialyloligosaccharides and sialylglycopeptides, prevent the adhesion of food borne pathogens including S. Enteritidis and *E. coli* to Caco-2 cell [81]. Another anti-adhesive fraction that was identified in egg yolks are high density lipoproteins (HDL), which have anti-adhesive activity in cell culture studies against S. Enteritidis and *Salmonella* Typhimurium [82]. Similarly, in the present study, glycoproteins in the eggshell cuticle could be anti-adhesive against *Salmonella*, as a negative correlation (R = −0.5472, *p* = 0.00176) was established between the ATR-FTIR sugar signal and bacterial adherence. A detailed future study is essential to determine the exact mechanism of interaction between bacterial pathogens such as Gram-positive *Bacillus cereus* and Gram-negative *Salmonella* and cuticle proteins. No large differences were observed in bacterial adhesion between eggs from different caging systems. However, data analysis showed that some significant differences (*p* = 0.01), with lower bacterial count in free-range as compared to enriched systems, while no difference was observed between conventional and alternative cages (Figure 7). This correlates with higher hydrophobicity of the eggshell surface from the free-range as compared to the enriched system. Additional biochemical analysis is required to elucidate the exact connection between the hydrophobicity and cuticle proteins, and their impact on the bacterial attachment to eggshell surfaces. The free-range alternative caging system offers the potential of better welfare when compared to other systems and enables normal bird behavior with reduced environmental stressors. This could modify the cuticle components on the eggshell surface (higher hydrophobicity or more abundant anti-adhesive proteins) to provide better protection against invading pathogens [7,83,84]. 

Overall, we did not observe large differences in the cuticle deposition, hydrophobicity, and bacterial adherence on eggshell surfaces between different housing systems. Good animal husbandry practices at all of the farms could be one reason for minor differences in the eggshell cuticle quality among the different caging systems [84,85,86]. Previous research has shown that animal handling by people can have a major impact on both the welfare and productivity of chickens. Studies have shown that poor egg production and quality is more prevalent on farms where the animals are roughly handled and are frightened of humans [87,88]; aversive handling causes animals to become more fearful, which results in physiological changes that typically occur when animals are stressed. This leads to deleterious effects on both animal welfare and productivity [85,89]. In addition, farm cleanliness, and the performance of various management routines are also major contributing factors to improve animal productivity [84]. Another reason for small differences in the cuticle quality and bacterial adherence between the housing systems could be that we sampled eggs from single farms within each housing system. In a future study, evaluation of differences within a given housing system from multiple farms would determine the true cuticle variability in the real production world. Additional studies are underway to address possible variability that are due to differences within housing systems at farm levels (with a target to evaluate eggs from multiple farms within barn types) to provide a clearer indication of which production system is associated with the best eggshell and cuticle quality. 

## 5. Conclusions

To conclude, this study validated our previous finding that the inner surface of the pore plug was consistently chemically different from the surrounding eggshell mineral composition in eggs from different housing systems. The cuticle and pore plugs were characterized using SEM and EDS and the chemical composition was determined using ATR-FTIR. The cuticle pores were longer, and the plugs were thicker in free-range as compared to conventional eggs. Eggs from alternative caging (enriched and free-range) had a higher total cuticle and a lower carbonate signal as compared to those from conventional cages. This information could be beneficial to poultry producers for transition from conventional to alternative caging as it will provide objective data for the selection of suitable caging systems. Better cuticle quality and lower bacterial adherence in alternative caging systems could be due to reduced environmental stresses in these housing systems. These results support our hypothesis that the housing systems can impact the cuticle parameters and therefore influence bacterial adherence. Further studies are required to identify hydrophobic/anti-adhesive cuticle components and evaluate the effect of the housing system on specific cuticle chemical components. We have reported, for the first time, that hydrophobicity can be used as a measure of the cuticle quality and predict the degree of bacterial adherence. These results can be used to develop new tools for predictive microbiology to maintain the food safety of table eggs. 

## Figures and Tables

**Figure 1 foods-10-02559-f001:**
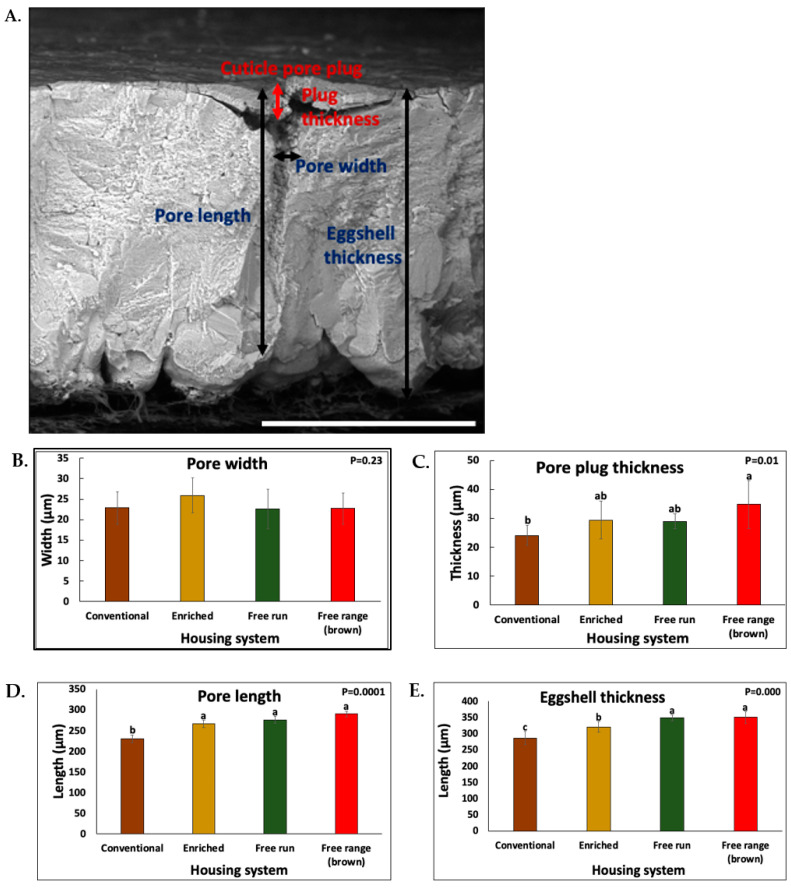
(**A**) The cross fractured eggshell surface showing the cuticle pore plugs as visualized by a scanning electron microscopy (SEM) at 1000×. The comparison of the eggshell cuticle pore plug parameters (**B**) Pore width, (**C**) Pore plug thickness, (**D**) Pore length, and (**E**) Eggshell thickness in eggs that were obtained from different housing system. SEM images (*n* = 18/housing system) were used to quantify cuticle pore plug parameters using the ImageJ software. The data shown corresponds to eggs from mid-lay hens. The values with different superscript letters (Tukey multiple means comparison) are significantly different (ANOVA; *p* < 0.05). The values represent the mean ± standard deviation from six independent experiments (*n* = 18/housing system).

**Figure 2 foods-10-02559-f002:**
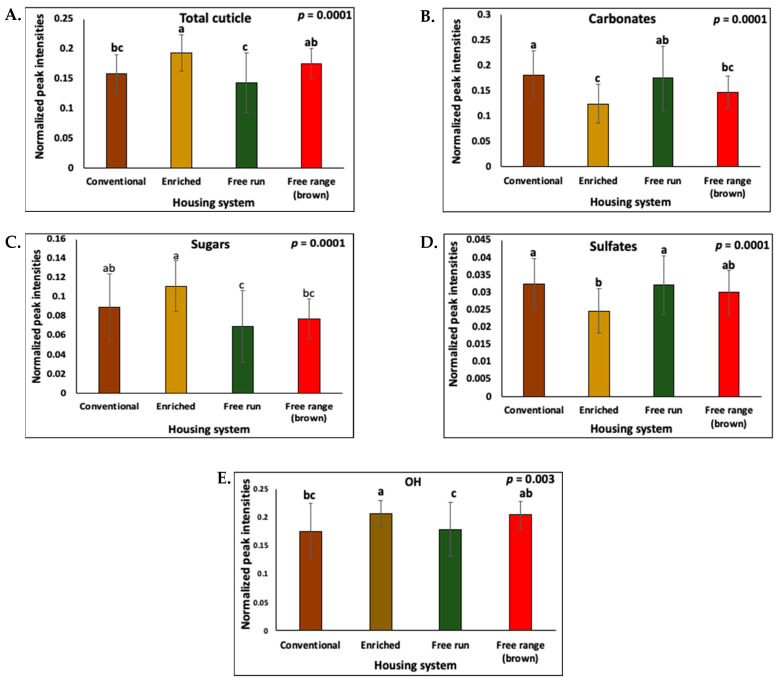
The intensity of ATR-FTIR spectroscopy peaks that were associated with the main components of the eggshell cuticle with the hen housing system (*n* = 30): (**A**) Total cuticle, (**B**) Carbonates, (**C**) Sugars/polysaccharides, (**D**) Sulfates, and (**E**) OH, (**F**) The table of correlation. The correlation between the cuticle chemical components of the eggshell cuticle that was established using a linear regression (Y_i_ = β_0_ + β_1_X_i_) in a curve fitting model using excel. The color coding indicates weak, moderate, or strong correlation between each chemical component of the cuticle. The values with different superscript letters (Tukey multiple means comparison) are significantly different (ANOVA; *p* < 0.05). The values represent the mean ± standard deviation (*n* = 30 eggs/housing system).

**Figure 3 foods-10-02559-f003:**
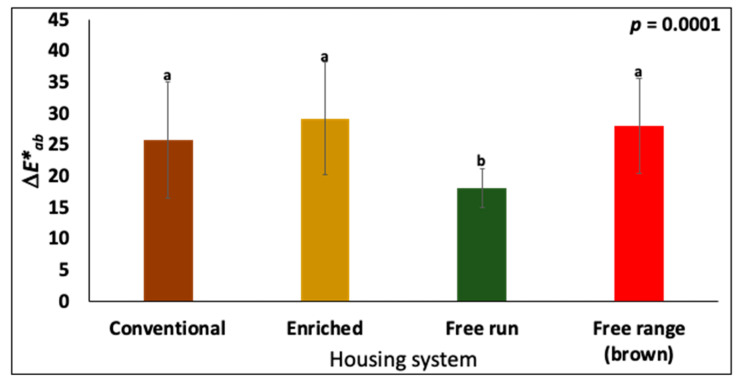
ΔE*_ab_ values of eggs from hens in different housing systems. The data shown corresponds to eggs from mid-lay hens. The values represent the mean ± standard deviation from each condition (*n* = 30). The values with different superscript letters (Tukey multiple means comparison) are significantly different (ANOVA; *p* < 0.05).

**Figure 4 foods-10-02559-f004:**
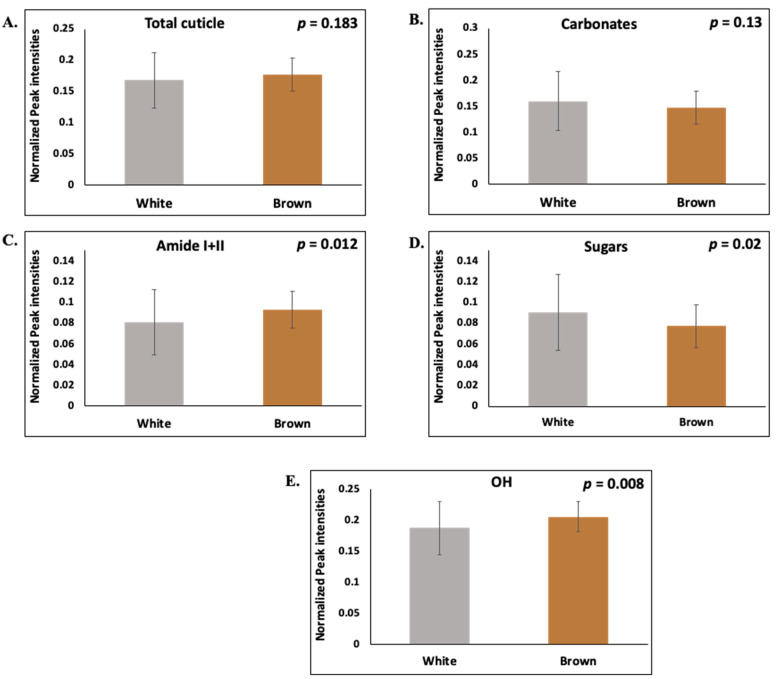
The hen strain dependence of the intensity of ATR-FTIR spectroscopy peaks that were associated with the main chemical components (**A**) Total cuticle, (**B**) Carbonates, (**C**) Amide I + II, (**D**) Sugars/polysaccharides, and (**E**) OH of the eggshell cuticle in white (*n* = 90) and brown (*n* = 30) eggs. The data shown corresponds to eggs from mid-lay hens. The values represent the mean ± standard deviation (Student’s *t*-test; *p* < 0.05).

**Figure 5 foods-10-02559-f005:**
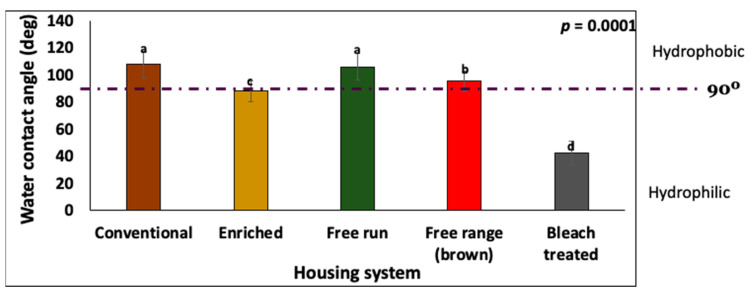
A comparison of the contact angle measurements on the eggshell surface between eggs that were derived from the different housing systems or bleach treated (no cuticle). The values with different superscript letters (Tukey multiple means comparison) are significantly different (ANOVA; *p* < 0.05). The values represent the mean ± standard deviation (*n* = 30).

**Figure 6 foods-10-02559-f006:**
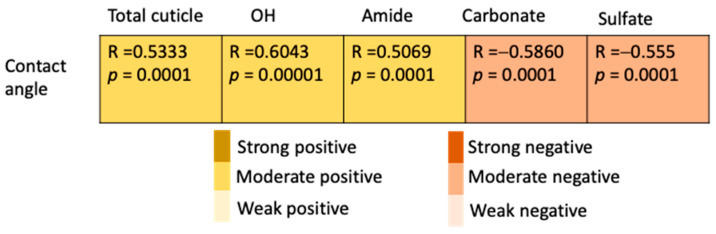
The correlation between the contact angle and the cuticle chemical components as quantified by AFT-FTIR spectroscopy. The correlations were established using a linear regression (Y_i_ = β_0_ + β_1_X_i_) in a curve fitting model using excel. The color coding determines a weak, moderate, or strong correlation between each chemical component of the cuticle.

**Figure 7 foods-10-02559-f007:**
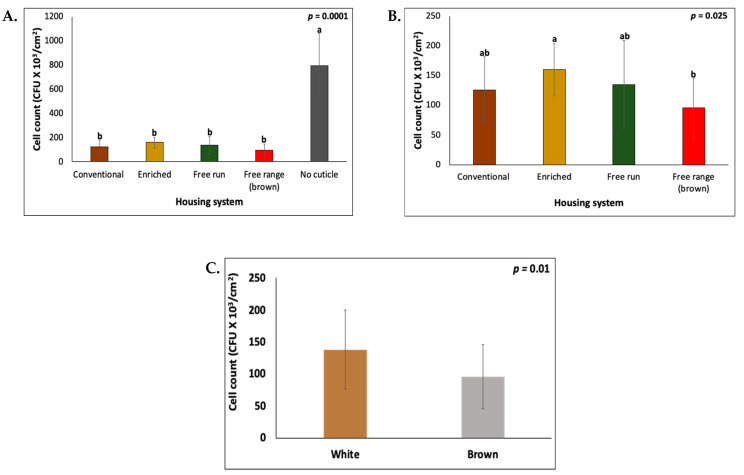
The bacterial cell attachment assay (**A**) The effect of the cuticle removal on *Salmonella* adherence on the eggshell surface. The *Salmonella* cell counts on the outer surface of the no cuticle (bleach-treated) eggshell were significantly higher than eggshells with an intact cuticle. (**B**) The impact of the different housing systems on *Salmonella* adherence to the eggshell surface. The values with different superscript letters (Tukey multiple means comparison) are significantly different (ANOVA; *p* < 0.05). The values represent the mean ± SD from six independent samples each performed in triplicate (*n* = 3). (**C**) The impact of the hen strain on *Salmonella* adherence to the eggshell surface. The values represent the mean ± standard deviation (*n* = 6 × 3 = 18, Student’s *t*-test; *p* < 0.05).

**Figure 8 foods-10-02559-f008:**
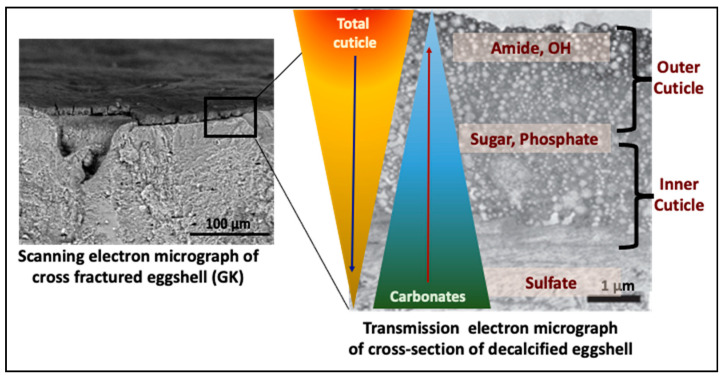
The proposed compositional gradient in the eggshell cuticle with predicted distribution of proteins, phosphoproteins, glycoproteins, and sulphated proteoglycans.

**Table 1 foods-10-02559-t001:** Comparison of the pore plug parameters in unwashed eggs of different colors: white and brown from hens at mid-lay.

Hen Strain	Pore Plug Parameters as a Function of Egg Color (Average ± SD)			
	Pore Width (μm)	Plug Thickness (μm)	Pore Length (μm)	Eggshell Thickness (μm)
White	23.81 ± 3.20	27.52 ± 4.38	257.49 ± 28.55 ^a^	318.30 ± 27.68 ^a^
Brown	22.57 ± 3.54	34.74 ± 7.50	290.52 ± 10.49 ^b^	349.49 ± 12.95 ^b^
*p*-value	0.471	0.06	0.000	0.0002

The values with different superscript letters (Tukey multiple means comparison) are significantly different (Student’s *t*-test; *p* < 0.05).

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
