# Peer review of "Impact of Different Layer Housing Systems on Eggshell Cuticle Quality and Salmonella Adherence in Table Eggs"

_foods, 2021, doi:10.3390/foods10112559_

Round 1

Reviewer 1 Report

Review of the Manuscript ID: foods-1413390 entitled Impact of different layer housing systems on eggshell cuticle quality and Salmonella adherence in table eggs.

The Manuscript submitted to me for review presents an extremely interesting and useful subject. Congratulations to the Authors of the idea as well as the method of carrying out the research. My suggestions are as follows:

General

First of all, the Authors throughout the Manuscript very often use slash, which in my opinion introduces illegibility. Rather, please replace it with and/or, and when You use a synonym, I would rather put the second word in bracket. This will organize the content and clarify the message.

Introduction

The Introduction presents important aspects emphasizing the need and “importance” of the proposed research, but I find this section too long. I recommend that You shorten it to about 1.5 typed pages.

Materials and Methods

Please, check company names and info… (some are missing, some are incomplete… The city is often missing)

Section 2.1. I believe that it would be better (for easier and more accessible reception and avoiding chaos) to separate the part on the source of eggs and description of housing systems (I) and the chemicals and instruments used in the study (II) into two separate subtitles.

Section 2.2. The abbreviations (eg. SEM) were first used in the Abstract and the full names are given there, so the abbreviations can be used in the following sections without the need to explain them.

Line 199: Please check the formula. There seems to be something missing... There is no brackets (at the end, isn’t it?)

Section 2.4. Wouldn't it be better to use a dash instead of a brackets? → Measurement of surface hydrophobicity – contact angle

Line 243: How was sonication performed? (apparatus) What were the parameters? You only entered the time

Results

Table 1: “Average± St Dev” Isn't it better: → Average ± SD

I have no comments on the description and presentation of the results.

Author Response

Reviewer 1:

The Manuscript submitted to me for review presents an extremely interesting and useful subject. Congratulations to the Authors of the idea as well as the method of carrying out the research. My suggestions are as follows:

  1. General

First of all, the Authors throughout the Manuscript very often use slash, which in my opinion introduces illegibility. Rather, please replace it with and/or, and when You use a synonym, I would rather put the second word in bracket. This will organize the content and clarify the message.

Response: We thank the reviewer for the constructive comments. The manuscript has been revised and use of slash is replaced with and / or /use of brackets.

  1. Introduction

The Introduction presents important aspects emphasizing the need and “importance” of the proposed research, but I find this section too long. I recommend that You shorten it to about 1.5 typed pages.

Response: We appreciate the suggestion of the reviewer; however, we feel that many important elements of the manuscript are well introduced in this section and prefer it in the current state.  

  1. Materials and Methods

Please, check company names and info… (some are missing, some are incomplete… The city is often missing)

Response: We thank reviewer for this valid comment. Missing information has been added to the method section.

  1. Section 2.1. I believe that it would be better (for easier and more accessible reception and avoiding chaos) to separate the part on the source of eggs and description of housing systems (I) and the chemicals and instruments used in the study (II) into two separate subtitles.

Response: As suggested by the reviewer, section 2.1 is separated into two sections:

  • Source of eggs and description of housing systems
  • Chemicals and instruments used in this study

  1. Section 2.2. The abbreviations (eg. SEM) were first used in the Abstract and the full names are given there, so the abbreviations can be used in the following sections without the need to explain them.

Response: As suggested by the reviewer, full name has been edited, and abbreviation of SEM has been used in the following sections.

  1. Line 199: Please check the formula. There seems to be something missing... There is no brackets (at the end, isn’t it?)

Response: We thank reviewer for this valid comment. Yes, a bracket is missing at the end. The formula has been edited.

  1. Section 2.4. Wouldn't it be better to use a dash instead of a brackets? → Measurement of surface hydrophobicity – contact angle

Response: As suggested by the reviewer, section heading has been modified.

  1. Line 243: How was sonication performed? (apparatus) What were the parameters? You only entered the time

Response: Sonication was performed for 5 min, 1 cycle at room temperature using an ultrasonicator, Fisher Scientific. This information has been added to the manuscript Line 258-260.

Reviewer 2 Report

The study is very comprehensive and deals with the question, if the thickness of the cuticle on the eggshell differs by housing systems. The authors consider the effect of different breeds (white vs brown layers), and report differences betwenn brown and white eggs, but also between housing systems. Eggs from free run / free range systems are obviously covered by a thicker cuticle and thus, better protected. Microtopography of the pores is explored. Attachment of Salmonella is explored and the detachmrnt of theses bacteria is studied (necessary for quantification of bacterial load). Whilst the observation that thicker shells have longer pores is somehow expected, the other observations contribute to our knowledge on the eggshell surface properties and on food safety of table eggs. Of particular interest is the finding that the contact angle measurement to measure hydrophobicity can be used to determine cuticle intactness.

There are some minor issues which should be addressed:

Abstract: the instructions for authors suggest 200 words, this abstract is much longer.

Line 64: „Battery cages are associated with..“ should be underpinned by a reference.

Line 106: SEM abbreviation should be explained here, at first mention, not in line 157.

Line 124: MST abbreviation should be explained here.Line 132: „furnished“ = „enriched“?

Line 115-156: the full name of ATR-FTIR has already been given, so the abbreviation/acronym would suffice.

Line 172: the different time of exposure to bleach should be explained. 10-60 min is a large variation.

Line 199: the formula is incomplete, at the end „)]“ is missing.

Line 229: the „8“ in „108“ must be superscript; similar for line 235

Line 232: „Salmonella“ in italics

Line 259: the P-value is here written with spaces, but when exact P is given, authors use no spaces. Maybe this should be written in a uniform way. (also in the footnotes to the figures)

Line 290 and others: Student´s

Line 360: „*IQR..“ means that the whiskers represent the IQR?

Lines 439,440: I think this can be expected anyway, but it is good to have it confirmed.

Line 533: „Bacillus cereus“ in italics

Lines 581,582: this paragraph must be completed.

Figure S4: the x-axis decription is misleading, although it is log-style, the numbers are not log, so: „bacterial dilution (1/ )“ would be better, or the axis description would be 1/100 ; 1/1000 etc. and the axis title: "bacterial dilution"

Author Response

Reviewer 2: The study is very comprehensive and deals with the question, if the thickness of the cuticle on the eggshell differs by housing systems. The authors consider the effect of different breeds (white vs brown layers), and report differences betwenn brown and white eggs, but also between housing systems. Eggs from free run / free range systems are obviously covered by a thicker cuticle and thus, better protected. Microtopography of the pores is explored. Attachment of Salmonella is explored and the detachmrnt of theses bacteria is studied (necessary for quantification of bacterial load). Whilst the observation that thicker shells have longer pores is somehow expected, the other observations contribute to our knowledge on the eggshell surface properties and on food safety of table eggs. Of particular interest is the finding that the contact angle measurement to measure hydrophobicity can be used to determine cuticle intactness. There are some minor issues which should be addressed:

  1. Abstract: the instructions for authors suggest 200 words, this abstract is much longer.

Response: We thank the reviewer for the constructive comments. The abstract is reduced to 200 words.

  1. Line 64: „Battery cages are associated with..“ should be underpinned by a reference.

Response: We thank reviewer for this valid comment. A reference has been added to Line 64.

  1. Line 106: SEM abbreviation should be explained here, at first mention, not in line 157.

Response: As suggested, SEM abbreviation has been added to Line 106.

  1. Line 124: MST abbreviation should be explained here.

Response: As suggested, the abbreviation has been explained in Line 124.

  1. Line 132: „furnished“ = „enriched“?

Response: Furnished cages has been designed to overcome hen welfare concerns in battery cages. As suggested, furnished cages include enriched cages in poultry production.

  1. Line 115-156: the full name of ATR-FTIR has already been given, so the abbreviation/acronym would suffice.

Response: The full name has been removed and abbreviation: ATR/FTIR is used in Line 115-156.

  1. Line 172: the different time of exposure to bleach should be explained. 10-60 min is a large variation.

Response: Previous literature including our study have shown that cuticle deposition on the outer surface of eggshell is not uniform and can be variable between eggs (Leleu et al., 2011; Rodriguez-Navarro et al., 2013; Bain et al., 2013). Moreover, hen age, commercial egg washing, hen strain, housing system can also affect cuticle deposition (Leleu et al., 2011; Samiullah et al., 2017; Kulshreshtha et al., 2018). Hence, time to remove cuticle was variable from one egg to another. We incubated each egg in bleach (5%) for 10 min and stained it with MST cuticle blue to confirm removal of the cuticle. This process was repeated up to 6 times (depending upon amount of cuticle on outer surface of each egg) in order to completely remove the cuticle. If outer eggshell surface stained green with MST cuticle blue then bleach treatment was repeated until complete removal of the cuticle. This information has been added to Line 172 for clarification.

- Rodriguez-Navarro, A.; Dominguez-Gasca, N.; Munoz, A.; Ortega-Huertas, M. Change in the Chicken Eggshell Cuticle with Hen Age and Egg Freshness. Poult Sci 2013, 92, 3026–3035.

-Leleu, S.; Messens, W.; de Reu, K.; de Preter, S.; Herman, L.; Heyndrickx, M.; de Baerdemaeker, J.; Michiels, C.W.; Bain, M. Effect of Egg Washing on the Cuticle Quality of Brown and White Table Eggs. Journal of food protection 2011, 74, 1649–54.

-Bain, M.; McDade, K.; Burchmore, R.; Law, A.; Wilson, P.; Schmutz, M.; Preisinger, R.; Dunn, I. Enhancing the Egg’s Natural Defence against Bacterial Penetration by Increasing Cuticle Deposition. Animal genetics 2013, 44, 661–668.

-Samiullah, S.; Omar, A.S.; Roberts, J.; Chousalkar, K. Effect of Production System and Flock Age on Eggshell and Egg Internal Quality Measurements. Poultry Science 2017, 96, 246–258.

- Kulshreshtha, G.; Rodriguez-Navarro, A.; Sanchez-Rodriguez, E.; Diep, T.; Hincke, M.T. Cuticle and Pore Plug Properties in the Table Egg. Poultry Science 2018, 97, 1382–1390.

  1. Line 199: the formula is incomplete, at the end „)]“ is missing.

Response: We thank reviewer for this valid comment. The formula has been modified.

  1. Line 229: the „8“ in „108“ must be superscript; similar for line 235

Response: As suggested, superscripts has been added to Line 229 and 235.

  1. Line 232: „Salmonella“ in italics

Response: As suggested, name of the bacteria has been italicised.

  1. Line 259: the P-value is here written with spaces, but when exact P is given, authors use no spaces. Maybe this should be written in a uniform way. (also in the footnotes to the figures)

Response: As suggested, P-value has been modified throughout the manuscript.

  1. Line 290 and others: Student´s

Response: Spelling of student’s t test has been modified throughout the manuscript.

  1. Line 360: „*IQR..“ means that the whiskers represent the IQR?

Response: We thank reviewer for this valid comment. Yes, as determined by statistical dispersion, interquartile range IQR is the difference between upper and lower quartiles. We used a well-established method based on “L*, a*, and b*” color space values to determine the intensity of green cuticle staining on the eggshell surface. The average of L*, a*, and b* values before (randomly selected) and after staining were calculated per eggshell fragment and used to calculate ΔE*ab (Leleu et al., 2011; Rodriguez-Navarro et al., 2013; Samiullah et al., 2013).

-Leleu, S.; Messens, W.; de Reu, K.; de Preter, S.; Herman, L.; Heyndrickx, M.; de Baerdemaeker, J.; Michiels, C.W.; Bain, M. Effect of Egg Washing on the Cuticle Quality of Brown and White Table Eggs. Journal of food protection 2011, 74, 1649–54.

- Rodriguez-Navarro, A.; Dominguez-Gasca, N.; Munoz, A.; Ortega-Huertas, M. Change in the Chicken Eggshell Cuticle with Hen Age and Egg Freshness. Poult Sci 2013, 92, 3026–3035.

-Samiullah, S.; and Roberts, J. The location of protoporphyrin in the eggshell of brown-shelled eggs. Poultry Science 2013, 92, 2783–2788., doi: 10.3382/ps.2013-03051.

  1. Lines 439,440: I think this can be expected anyway, but it is good to have it confirmed.

Response: We agree with this comment. Thank you.

  1. Line 533: „Bacillus cereus“ in italics

Response: As suggested, name of the bacteria has been italicised.

  1. Lines 581,582: this paragraph must be completed.

Response: As suggested, the paragraph has been completed.

  1. Figure S4: the x-axis decription is misleading, although it is log-style, the numbers are not log, so: „bacterial dilution (1/ )“ would be better, or the axis description would be 1/100 ; 1/1000 etc. and the axis title: "bacterial dilution"

Response: We thank reviewer for this valid comment. As suggested, the graph has been modified for better understanding.
